# Comparative Study on Phytochemical Profile and Antioxidant Activity of an Epiphyte, *Viscum album* L. (White Berry Mistletoe), Derived from Different Host Trees

**DOI:** 10.3390/plants10061191

**Published:** 2021-06-11

**Authors:** Mahak Majeed, Tanveer Bilal Pirzadah, Manzoor Ahmad Mir, Khalid Rehman Hakeem, Hesham F. Alharby, Hameed Alsamadany, Atif A. Bamagoos, Reiaz Ul Rehman

**Affiliations:** 1Department of Bioresources, University of Kashmir, Hazratbal 190006, India; forestryupm@gmail.com (M.M.); bioinformatics.e@gmal.com (M.A.M.); 2University Centre for Research and Development (UCRD), Chandigarh University, Punjab 140413, India; pztanveer@gmail.com; 3Department of Biological Sciences, Faculty of Science, King Abdulaziz University, Jeddah 21589, Saudi Arabia; halharby@kau.edu.sa (H.F.A.); halsamadani1979@hotmail.com (H.A.); abamagoos@kau.edu.sa (A.A.B.)

**Keywords:** antioxidant, extraction, epiphyte phytochemical, plant extract

## Abstract

The study aimed at evaluating the antioxidant profile of a medicinal epiphyte *Viscum album* L. harvested from three tree species, namely, *Populus ciliata* L, *Ulmus villosa* L., and *Juglans regia* L. The crude extracts were obtained with ethanol, methanol, and water and were evaluated for the total phenol content (TPC), total flavonoid content (TFC), and antioxidant activities using total reducing power (TRP), ferric reducing antioxidant power (FRAP), 1, 1-diphenyl 1-2-picryl-hydrazyl (DPPH), superoxide radical scavenging (SOR), and hydroxyl radical scavenging (^•^OH) assays. Our results showed that crude leaf extracts of plants harvested from the host *Juglans regia* L. exhibited higher yields of phytochemical constituents and noticeable antioxidative properties. The ethanolic leaf samples reported the highest phenols (13.46 ± 0.87 mg/g), flavonoids (2.38 ± 0.04 mg/g), FRAP (500.63 ± 12.58 μM Fe II/g DW), and DPPH (87.26% ± 0.30 mg/mL). Moreover, the highest values for TRP (4.24 ± 0.26 μg/mL), SOR (89.79% ± 0.73 mg/mL), and OH (67.16% ± 1.15 mg/mL) were obtained from aqueous leaf extracts. Further, Pearson correlation was used for quantifying the relationship between TPC, TFC, and antioxidant (FRAP, DPPH, SOR, OH) activities in *Viscum album* L. compared to their hosts. It was revealed that the epiphyte showed variation with the type of host plant and extracting solvent.

## 1. Introduction

Over the ages, plants have been known as vital natural reservoirs of secondary metabolites and as such, extensive efforts are being directed toward the research and development of phytomedicines comprising flavonoids, vitamins, alkaloids, tannins, and terpenoids as key therapeutic agents for the treatment of various metabolic diseases that have been ascribed to their indispensable biological activities such as detoxification of toxic enzymes, inhibition of cellular damage, regulation of gene expression, and antimicrobial and anti-inflammatory actions [1,2,3]. Several lines of research suggest that ingestion of natural antioxidants can reduce the risk of various health complications including cancers, neurodegenerative disorders, and diabetes [4,5]. Most of the beneficial roles played by natural antioxidants in maintaining human health resulted from their reducing potential in quenching free radicals such as reactive oxygen or nitrogen species (ROS/RNS) due to their hydrogen donating tendency and thus preventing the oxidative damage of cells caused by the action of free radicals [6]. Free radicals are very reactive chemical species, noticeably hydroxyl radicals (^•^OH) and superoxide ion (O_2_^−^) that react with important biological compounds such as phospholipids, proteins, and nucleic acids, leading to oxidative damage in healthy cells of the body [7]. Recent studies have demonstrated that the defensive role of antioxidants against oxidative stress is initiated by regulating the expression and activity of key proteins involved in intracellular signaling pathways through their binding interaction with antioxidants [8,9]. Other mechanisms shown by antioxidants to counteract oxidative stress are mediated by regulating the activity of gut microbiota or modulating epigenetics [5,10,11,12,13]. Various studies have confirmed the inhibitory role of natural antioxidants present in plant extracts under in vitro conditions against the harmful effects induced by free radicals [14]. Nowadays, there is also growing interest in natural therapeutics due to their lower toxicities than their chemical counterparts, which have a wide application in medicine, food, and cosmetic industries [15,16]. *Viscum album* L. or common European Mistletoe from the Santalaceae family is a potential medicinal evergreen shrub that grows on diverse kinds of woody trees due to its semi-parasitic mode of nutrition [17]. The epiphyte exhibits photosynthetically active leaves that aid in the synthesis of its compounds, hence showing its hemiparasitic mode of nutrition. However, for nutrient supply, this species strictly relies on the xylem sap of the host plant for extracting some pharmacologically significant metabolites, but due to a lack of connection between the phloem tissues of the host and epiphyte, the flow of photo- assimilates is restricted from the epiphyte to the host [18,19]. European Mistletoe is well known for various pharmacological effects against various stress-related disorders including diabetes, hypertension, epilepsy, arthritis, and cancer [20]. Furthermore, the phytochemical investigation conducted on the plant revealed some important therapeutic agents, namely, viscin, viscotoxin, saponins, flavonoids, acetylcholine, lectins, mucilage ascorbate, and tocopherol [21,22,23].

In this study, we investigated the phytochemical and antioxidant potential of stems, leaves, and berries of *Viscum album* L. harvested from *Populus ciliata* L., *Ulmus villosa* L., and *Juglans regia* L. and extracted in three solvents: ethanol, methanol, and water. The variations in the phytochemical yields and antioxidative potential of *Viscum album* L. were investigated in relation to different host trees.

## 2. Materials and Methods

### 2.1. Plant Material Collection and Processing

The plant collection was conducted in Ferozpora areas in Tangmarg town in the district of Baramulla of Jammu and the Kashmir region during the winter season, December 2018, and plant materials (stem, leaf, and berry) were collected from three different host trees, namely, *Populus ciliata* L., *Ulmus villosa* L., and *Juglans regia* L., (Figure 1). The plant material was sorted according to the parts of the epiphyte and type of host tree; the material was then air dried in the shade for almost two months. The air-dried samples were ground into a fine powder using an electric grinder, and the powdered plant material was stored in labelled airtight amber bottles for antioxidant analysis.

### 2.2. Extraction Procedure

Double distilled water, methanol (80%), and ethanol (80%) were chosen as extraction solvents based on their significant antioxidant values as observed in studies with other medicinal plants. Finely ground powdered plant material (5 g) was extracted in the above-mentioned solvents. The mixture was kept on an incubator rotatory shaker at (200 rpm) 25 °C for 48 h to preserve the natural antioxidants in a sample extract and for allowing maximum extraction. Extracts were then filtered through Whatman filter paper (No. 1), and the filtrate was centrifuged at 8000 rpm at 12 °C for 15 min to obtain a clear supernatant. The extracts obtained in different solvents were evaporated under reduced pressure in a rotatory evaporator and were rediluted to obtain the concentration of 10 mg/g as a stock solution that was stored in falcon tubes wrapped with aluminium foil at 4 °C in the refrigerator for further use. The extraction yield was calculated using the following equation.
Extraction yield (%) = Amount (g) of dried crude extract obtained/Amount (g) of finely powdered plant material × 100

### 2.3. Phytochemical Profiling

#### 2.3.1. Total Phenolic Content (TPC)

TPC of extracts was determined by Folin–Ciocalteu reagent following Idris et al. [24] with some modifications and using gallic acid as a standard phenolic compound. Plant extracts were taken in 0.1 mg/µL concentrations, which were added to diluted FC reagent. After a few minutes, 20% Na_2_CO_3_ was added to the reaction mixture, and the reaction mixture was shaken well before placing in a boiling water bath for 2 min. The absorbance was measured spectrophotometrically at 765 nm, and the TPC was determined as mg of gallic acid equivalents (GAE) per gram of crude extract deduced from the standard linear graph y = 0.0081x + 0.0183, R^2^ = 0.9992.

#### 2.3.2. Total Flavonoid Content (TFC)

TFC was evaluated following aluminium chloride (AlCl_3_) spectrophotometric assay according to Ohikhena et al. [25] with few modifications. In this assay, plant extracts at 0.1 mg/µL concentration were added to 0.5 mL of AlCl_3_ (0.1 M) and 2 mL of potassium acetate (CH_3_COOK) (1 M). The reaction tubes were incubated at room temperature for 30 min. The optical density (OD) was measured at 415 nm, and the compound rutin was used as a standard flavonoid, and total flavonoids were evaluated as mg/g of rutin equivalent (RE)/g of crude extract obtained from the calibration curve, y = 0.001x + 0.0058, R^2^ = 0.987.

#### 2.3.3. Antioxidant Activity based on the Total Reducing Power (TRP), Ferric Reducing Antioxidant Power(FRAP), 1, 1-diphenyl 1-2-picryl-hydrazyl (DPPH), Superoxide Radical Scavenging (SOR), and Hydroxyl Radical Scavenging (OH-) Assays

TRP of plant extracts was determined by measuring the tendency of plant extracts to reduce FeCl_3_ as described by Moein et al. [26] with few modifications. Appropriate dilutions of plant extracts in 25 µL, 50 µL 75 µL and 100 µL were mixed with 0.5 mL of 0.2 M of Na_3_PO_4_ buffer at 6.6 pH and 0.5 mL of 1% potassium ferricyanide (K_3_FeCN_6_). The mixture was incubated at 50 °C for 20 min. Following cooling, 0.5 mL of 10% trichloroacetic acid (TCA) was added to the extract mixture, which was centrifuged at 2500 rpm for 10 min. Then, 0.5 mL of supernatant was mixed with an equal volume of distilled water, and 100 µL of 0.1% ferric chloride (FeCl_3_) was added. The absorbance of the plant extract mixture was measured at 700 nm using spectrophotometer (Shimadzu UV-1800, Kyoto, Japan) against a blank reagent. Higher values showed a higher reducing potential of plant extracts.

FRAP assay was followed according to the method proposed by Pang et al., [27] with few modifications. Briefly, 300 mM sodium acetate (CH_3_COONa) buffer solution at pH 3.6, 10 mM TPTZ (2, 4, 6-tripyridyl-s-triazine) solution, and 20 mM ferric chloride (FeCl_3_) solution were prepared and mixed in the ratio of 10:1:1.The FRAP reagent was incubated at 37 °C before use, and the plant extracts were mixed with 1.9 mL FRAP reagent. After incubation for 10 min at 25 °C, the absorbance was measured at 593 nm. The FRAP values were calculated and expressed in terms of the dry weight of the samples as μM of ferrous equivalent Fe (II) per gram of sample.

DPPH assay was performed following the methodology used by Nickavar et al. [28] with slight modification. In this assay, a 0.1 mM solution of DPPH radicals was prepared by dissolving 39.4 mg of DPPH in 100 mL ethanol. Then, 2 mL of ethanolic solution of DPPH was added to plant extracts in 20 µL, 30 µL, 40 µL, and 50 µL concentrations. The reaction mixture was then allowed to stand at room temperature for 30 min, and absorbance was recorded at 518 nm. The percentage of DPPH scavenging was then calculated using ultraviolet-visible (UV/VS) Shimadzu spectrophotometer with the following equation:DPPH scavenged (%) = [(A_control_ − A_sample_)/A_control_] × 100
where A_control_ is a solution containing 3 mL of ethanolic DPPH solution without plant extracts. A_sample_ is the reaction mixture containing plant extracts, and ethanol (3 mL) was used as a blank.

The superoxide radical scavenging assay was performed following the method of Liu and Ng [29] with slight modification. The radicals were generated under in vitro conditions in 16 mM Tris-HCl buffer at pH 8 that contained 78 µM β- nicotinamide adenine dinucleotide (NADH), 10 µM phenanzine methosulphate (PMS) 50 µM nitroblue tetrazolium (NBT), and plant extracts in 20 µL, 30 µL, 40 µL and 50 µL concentrations The reactions between NBT and SOR radicals were assayed by the development of purple formazan colour measured spectrophotometrically at 560 nm. However, the addition of plant extract in the reaction mixture quenches superoxide radicals O_2_, leading to inhibition of NBT reduction. This was shown by a lower absorbance of the reaction mixture and hence higher SOR scavenging. The SOR inhibition percentage was calculated using the following equation:SOR scavenging (%) = [(A_control_ − A_sample_)/A_control_] × 100
where A_control_ is the absorbance of the reaction mixture without plant samples while A_sample_ is the absorbance after reduction of NBT radicals by plant samples.

The hydroxyl radical scavenging potential was evaluated by the salicylate method proposed by Zhao et al. with slight modifications [30]. Briefly, 4 mL of reaction mixture containing 1 mL plant extracts in 20 µL, 30 µL, 40 µL and 50 µl concentrations was mixed with 1 mL each of salicylic acid dissolved in absolute ethanol (9 mM), FeSO_4_ (6 mM), and H_2_O_2_ (24 mM). The reaction was initiated by the addition of H_2_O_2,_ and the reaction mixture was incubated at room temperature for 30 min. The absorbance was measured at 510 nm. The OH radical scavenging percentage was calculated using the following equation:OH % scavenging = [(A_control_ − A_sample_)/A_control_] × 100
where A_control_ is the absorbance of the control without samples, and A_sample_ is absorbance after adding H_2_O_2_ to the reaction mixture.

### 2.4. Statistical Analysis

All the experiments were carried out in triplicate, and the results derived were calculated as mean ± standard errors (SEs) of three parallel readings. Significant differences between samples were analysed by two-way analysis of variance (ANOVA) with Tukey test of multiple comparisons using GraphPad prism 6. The *p* values < 0.05 were considered statistically significant. IC_50_ values were deduced by analysing the linear regression equation using MS Excel 2010. The Pearson correlation coefficient and heatmap were used to evaluate the relationship between the antioxidant activity and the phenolic and flavonoid contents in the samples (MetaboAnalyst software version 5.0). Likewise, this coefficient and its significance level were resolved to illustrate how the biological host–parasite relationship influences the polyphenol content and antioxidant activity.

## 3. Results

### 3.1. Effect of Host Plants on the Phytochemical Content

Hemiparasitic plants (Mistletoe) have been investigated for the absorption of nutrients, water, and metabolites from their respective hosts [31,32], and it has been revealed that the metabolic profile of Mistletoe changes significantly with the type of host plant, which affects its antioxidant activity [33,34,35] and could be ascribed to differences in polyphenolic contents [36]. For determining the antioxidant potential of the plant extracts, there is no specific evaluation method due to the structural diversity of compounds, their modes of action, and multiple patterns of interactions [37]. The analytical accuracy of any given antioxidant assay depends on its sensitivity, selectivity, and linearity [38]. Most accurate results can be obtained from antioxidant assays when applied to the appropriate problems [39]. Therefore, the total antioxidant potential of samples can be ascertained by application of different and most suitable antioxidant assays involving different working principles [40].

#### 3.1.1. Total Phenolic Content (TPC)

The TPC in different parts of the plant was evaluated as mg/g of gallic acid equivalent (GAE) as indicated in (Figure 2A) below. Methanolic and ethanolic sample extracts recorded higher phenolic contents. The ethanolic berry and methanolic stem and leaf extracts of *Viscum album* L., hosted by *Populus ciliata* L., recorded the highest phenolic compositions as 19.7 ± 0.38 mg/g, 17.6 ± 0.17 mg/g, and 13.87 ± 0.03 mg/g, respectively. Further, the phenolic content was higher in the methanolic and ethanolic leaf samples harvested from host *Juglans regia* L., as 13.46 ± 0.87 mg/g and 13.73 ± 0.83 mg/g. The lowest values of phenols, 2.44 ± 0.05 mg/g and 2.52 ± 0.25 mg/g, were measured in aqueous and ethanolic berry extracts from the host *Ulmus villosa* L.

#### 3.1.2. Total Flavonoid Content (TFC)

Total flavonoids were estimated as mg/g of plant extract in rutin equivalents as shown (Figure 2B). Similar to the results obtained with phenolic contents, the ethanolic and methanolic samples extracted more flavonoids than aqueous samples. The ethanolic extracts of the leaf obtained from hosts *Ulmus villosa* L. and *Juglans regia* L. showed the highest yields of flavonoids with values measured as 2.61 mg/g ± 0.15 and 2.38 ± 0.04 mg/g, while aqueous berry extracts had the flavonoid values that ranged from 0.41 ± 0.01 to 0.36 ± 0.01 (RE)/g of crude extracts of berry samples harvested from hosts *Juglans regia* L. and *Ulmus villosa* L., respectively.

#### 3.1.3. Total Reducing Power (TRP) Assay

The total antioxidant activity of different sample extracts was estimated with the total reducing power (TRP) assay as depicted (Figure 2C). The aqueous leaf extract, obtained from host tree *Juglans regia* L., showed the highest reducing potential with values measured as 4.24 ± 0.26 μg/mL compared with other extract samples evaluated for the assay. The *Ulmus villosa* L.-derived methanolic berry samples also exhibited higher reducing capacity, with values of 2.84 ± 0.17 μg/mL. However, the lowest activity was noted in aqueous stem extracts obtained from the host plants *Populus ciliata* L. and *Ulmus villosa* L., with total reducing power values that ranged from 0.56 ± 0.02 μg/mL to 0.34 ± 0.01 μg/mL, respectively.

The ferric reducing antioxidant (FRAP) assay was conducted to measure the reducing potential of antioxidants present in plant extracts against the oxidative effect of reactive oxygen species (Figure 2D). Among various plant samples derived from different host plants, the maximum reducing capacity was reported in all leaf samples obtained from the host *Juglans regia* L., most noticeably in ethanolic leaf extracts with the highest ferric reducing potential of 500.63 ± 12.58 μM Fe II/g DW. Aqueous extracts showed a reduced ferric reducing potential. The lowest FRAP values calculated were 50 ± 0.60 μM Fe II/g DW and 56.45 ± 0.22 μM Fe II/g DW, recorded in aqueous stem and leaf extracts, respectively, obtained from the host plant *Ulmus villosa* L.

#### 3.1.4. DPPH Assay

The DPPH radical scavenging assay reported the variation in the scavenging potential of extracts derived from various host plants and extracted with various solvents. According to results derived from this assay, the different extracts showed solvent- and concentration-dependent scavenging potential. Methanolic and ethanolic extracts showed comparatively higher scavenging activity, with the highest values of 93.8% ± 1.24 mg/mL and 90.28% ± 0.45 mg/ml measured in methanolic berry and stem samples from hosts *Populus ciliate* L. and *Ulmus villosa* L. (Figure 3A,C). Moreover, higher scavenging values of 90.03% ± 0.45 mg/mL, 90.03% ± 0.13 mg/mL, and 87.26% ± 0.30 mg/mL were also reported in the methanolic stem, ethanolic stem, and leaf extracts harvested from host *Juglans regia* L. However, the aqueous extracts had lower radical scavenging activities with the lowest values of 23.66% ± 2.51 mg/ml and 32.21% ± 1.50 mg/ml determined mostly in aqueous leaf and stem extracts, respectively, derived from host *Ulmus villosa* L. (Figure 3A,B).

Further results were examined from different series of sample concentrations to determine the IC_50_, the concentration of plant extracts at which fifty percent of DPPH radicals were scavenged. The lowest IC_50_ measurements, 0.024 μg/mL and 0.025 μg/mL, were reported in methanolic and ethanolic stem extracts from hosts *Ulmus villosa* L. and *Juglans regia* L., respectively (Table 1).

#### 3.1.5. SOR Assay

The highest SOR scavenging percentages were detected in the aqueous and methanolic leaf extracts with values of 94.36% ± 0.47 mg/mL and 90.92% ± 1.48 mg/mL harvested from host plants *Ulmus villosa* L. and *Juglans regia* L., respectively (Figure 4B). Moreover, the aqueous stem and leaf samples obtained from *Juglans regia* L. also depicted high SOR reducing percentages of 90.87% ± 1.13 mg/mL and 89.79% ± 0.73 mg/mL (Figure 4A,B). The lowest SOR scavenging percentages of 29.44% ± 5.39 mg/mL and 30.26% ± 2.0 mg/ml were observed in methanolic and ethanolic berry extracts derived from hosts *Populus ciliata* L. and *Ulmus villosa* L., respectively (Figure 4C).

The IC_50_ data of SOR absorbing capacities of plant samples obtained from three host plants are presented (Table 1). LowerIC_50_ values, 0.30 μg/mL and 0.34 μg/mL were recorded in aqueous leaf and ethanolic berry samples, respectively, from host *Juglans regia* L. The ethanolic stem extract derived from host *Ulmus villosa* L. had the lowest IC50 value, 0.09 μg/mL.

#### 3.1.6. ^•^OH Assay

The highest OH radical reducing percentages were 75.81% ± 1.11 mg/mL and 68.91% ± 0.67 mg/mL in methanolic leaf extracts harvested from hosts *Populus ciliata* L. and *Ulmus villosa* L., respectively. According to the results, significant OH scavenging percentages were also measured in aqueous leaf extracts and methanolic stem extracts from hosts *Juglans regia* L and *Populus ciliata* L., with values of 67.16% ± 1.15 mg/mL and 61.93% ± 2.8 mg/mL (Figure 5A,B). It is also noteworthy to mention that berry extracts had a lower OH reducing potential, while the lowest values of 4.06% ± 2.22 mg/mL and9.19% ± 2.86 mg/ml were detected in berry extracts from hosts *Juglans regia* L. and *Populus ciliata* L., respectively (Figure 5C). The lowest IC_50_ values were calculated in stem and leaf extracts from host *Populus ciliata* L. as methanolic stem > methanolic leaf > ethanolic leaf extract (Table 1). Thus, this assay confirmed that the ethanolic leaf extract from host *Populus ciliata* L. with the lowest IC_50_ value of 0.74 μg/mL was the most potent in reducing OH radicals among extracts examined for the assay.

#### 3.1.7. Relationship between Epiphyte–Host Plants in Total Phenolic, Flavonoid, and Antioxidant Activity

Pearson’s correlation coefficient was used for quantifying relationships between various parts of the epiphytic plant *Viscum album* L. and its different hosts. Figure 6 shows the Pearson correlations and levels of significance for the relationship between the total phenolic content (TPC), total flavonoid content (TFC), and antioxidant (FRAP, DPPH, SOR, OH) activities between the host trees and the various parts of the epiphytes. The results of the different antioxidant assays used in the present investigation of different parts of the *Viscum album* L. extracts were compared and correlated with each other. Correlation between the results of different antioxidant assays is presented in Figure 7. The total phenolic content (TPC) showed a good correlation with most of the antioxidant assays, such as FRAP (R^2^ = −0.520*), SOR (R^2^ = −0.192), and DPPH radical scavenging assay (R^2^ = −0.448) (Table 2). Similarly, the total reducing power (TRP) exhibited a moderate correlation with hydroxyl radical (OH) scavenging activity. Further, the total flavonoid content (TFC) showed a correlation with DPPH radical scavenging activity (R^2^ = −0.448). Considering all *Viscum*-host pairs, there were several remarkable differences. There were clear tendencies for *Viscum album* L. and its hosts to differ in total phenols, with *Viscum album* L. showing a higher polyphenol content among the five pairs. A a strong correlation for the total reducing power (TRP) and hydroxyl radical (OH) scavenging activity was found between the *Viscum album* L. leaf and its host *Juglans regia* L. followed by *Viscum album* L. berry and its host *Ulmus villosa* for total reducing power (TRP) (R^2^ = 0.636**), while least significant correlations were detected for the other pairs.

## 4. Discussion

The medicinal significance of a plant is attributed to its plethora of phytoconstituents [41]. These phytoconstituents, including alkaloids, phenols, and flavonoids, play diverse biological roles in plant–pollinator interactions, reactive oxygen scavenging, plant defence, and metal chelation [42,43,44,45,46]. There is a marked effect of different host plants on the relative concentration and activity of the key phytochemicals. The nutrient balance of plants is governed by an intricate combination of an extrinsic nutrient supply and intrinsic nutrient trade-off that affects the synthesis and storage of defensive phytochemicals in plant tissues [47]. These attributes raise interest in evaluating the antioxidant variation resulting from the difference in abundance of phytochemicals in *Viscum album* L. harvested from different host plants in the determination of the type of host plant species with maximum photochemical yield and thus greater antioxidant potential. In contrast to aqueous extract, ethanolic and methanolic solvents had higher yields of phenolic contents. This can be explained due to the difference in polarity and eluent strength among ethanol, methanol, and water. Ethanol and methanol are best suited to extract compounds displaying a wide range of polarities, while water has been reported to be suitable only for extracting highly polar compounds [48]. Besides the nature of the solvent, another important factor influencing the solubility of phenols was found to be the chemical structure of phenolic compounds [49]. According to studies conducted on biological activities of phenolic compounds, phenols play a significant role in antioxidant activity by quenching free radicals, singlet oxygen (O_2_^−^) or metal ions (Fe^2+^) due to their lower redox potential [50,51]. The strong correlation between total phenolic contents of a plant and their resultant antioxidant properties has been well supported by various antioxidant studies conducted on plant extracts [52,53,54,55,56,57,58]. Similar to findings reported in phenolic compositions, the total flavonoid contents were also highest in ethanolic and methanolic extracts. Flavonoids represent the most ubiquitous phytochemicals in the human diet. These are easily absorbed via gut epithelium, transported in blood plasma, and excreted in the urine. Flavonoids are associated with diverse biological functions such as free radical scavenging, metal chelating activity, cardio-protective, and hepatoprotective, anti-inflammatory, and antitumor activities [59,60]. The results obtained for the total phenols and flavonoid contents of *Viscum album* L. for different host plants justify that plant organ and host species strongly influence the phytochemical yields and thus the antioxidant activities of an epiphyte. These observations are in accordance with the studies conducted by Vicas et al., and Urech et al. [61,62]. Our study also highlighted the variation in total yields of phenols and flavonoids in different extracting solvents that can be explained by the difference in the availability of extractable compounds arising due to the formation of different types of complexes formed from these compounds with other phytochemicals in various samples of plant material. These findings are also justified by Bushra et al. and Hsu et al. [63,64], who investigated the effect of different extracting solvents on the antioxidant potential of medicinal plants. The significant indicator of the antioxidant assay was evaluated through TRP assay, which is based on the hydrogen donating tendency of plant extracts. The TRP assay revealed the highest reducing potential in aqueous leaf and methanolic berry samples obtained from hosts *Juglans regia* L. and *Ulmus villosa* L., respectively. These samples showed the highest reducing potential and can act as potent electron donors in reducing the free radicals and thus neutralising them into stable products causing an interruption in the chain reaction [65].The results of reducing the potential of mistletoe further confirm that the total antioxidant properties of *Viscum album* L. significantly vary with the type of its host tree [66,67,68]. It is important to mention that the antioxidant activity shown by plant extracts is also dependent on the type of extracting solvent. It has been well documented that the nature of extracting solvents affects both the yield of phytochemical constituents and hence their cumulative antioxidant activities. This can be explained by the high variability in chemical characteristics and polarities of phytoconstituents that lead to differences in their solubilities in different solvents [63,69]. Moreover, the significant differences in the values of total reducing power between *Juglans regia* L.-derived aqueous leaf extracts and other samples of Mistletoe can be ascribed to the heterogeneous nature of antioxidants and also the environmental factors influencing the synthesis and storage of antioxidants in parts of the plant [70,71,72].

Moreover, the radical scavenging potential of *Viscum album* L. harvested from various hosts was evaluated through FRAP, DPPH, SOR, and OH assays. DPPH assay is considered the most significant test to evaluate the free radical absorbing capacity of plant extracts. The DPPH radical scavenging potential of plant extracts results mainly from their hydrogen donating tendency [73]. Various research studies documented that the high phenolic content of the extracts was also correlated with their significant radical scavenging abilities. The phenolic contents in plant extracts are responsible for scavenging free radicals. These free radicals are associated with the generation of various chronic diseases including cancers [74,75,76]. It is also worth mentioning that free radicals such as superoxide anion radicals (O_2_^−^), hydroxyl radical (OH^−^), alkoxyl radical (RO^−^), and peroxyl radical (ROO^−^) are known to play an indispensable role in the normal metabolism of an organism at the cellular level [77]. The loss of equilibrium between levels of antioxidants and free radicals is characterized by oxidative stress that leads to the generation of various chronic diseases [78]. In FRAP and DPPH assays, higher ferric and DPPH+ reducing potential was observed in methanolic and ethanolic extracts from all host samples, which also showed significant TPC values. These findings are also in agreement with studies conducted by Simona et al., on methanolic and ethanolic extracts of *Viscum album* L. [61]. Besides this, the higher antioxidant potential of methanolic and ethanolic extracts observed in FRAP and DPPH assays can be due to the high solubility potential of ferric and DPPH radicals in methanol and ethanol due to which these radicals follow the same mechanism in transferring hydrogen [79]. The superoxide radical scavenging potential of Mistletoe extracts was evaluated by the NBT method. The superoxide radicals are known to be highly reactive oxygen species that are released from the cell as a response to normal aerobic metabolism. Studies have suggested that SOR triggers the formation of more lethal reactive oxygen species in the form of hydrogen peroxides (H_2_O_2_) and hydroxyl radicals (OH^•^). These radicals were found to impair the normal functions of the cell by damaging its lipids, proteins, and nucleic acids and thus inducing oxidative damage [80]. The results showed that superoxide radicals were inhibited regardless of the nature of extracting solvent. According to our results, the aqueous leaf extracts from host *Ulmus villosa* L. showed the highest superoxide radical scavenging potential that is further justified by its lowest IC_50_ value measured as 0.84 μg/mL. The IC_50_ data of the SOR assay also revealed that ethanolic leaf samples from host *Ulmus villosa* L. can reduce the superoxide radicals to the lowest concentration of 0.09 μg/mL of extract. Moreover, the lowest IC_50_ values were also reported in aqueous leaf extract from host *Juglans regia* L., measured as 0.31 μg/mL. Therefore, the results evaluated for the SOR assay suggested that ethanolic and aqueous leaf extracts are the best scavengers for superoxide radicals (O_2_^−^) that are released in PMS-NADH-NBT systems in vitro. The positive SOR reducing capacity of aqueous and ethanolic leaf extracts can be ascribed to the presence of phenols and flavonoids [81]. In the ^•^OH assay, the highest ^•^OH scavenging potential was observed in methanolic leaf extracts obtained from hosts *Populus ciliata* L. and *Ulmus villosa* L. The ^•^OH radical is known as a highly reactive radical with a short half-life of ~10^─^^9^ s [82]. Being the most harmful oxygen-carrying free radical, it attacks fundamental biological molecules of the cell such as proteins, nucleic acids, and membrane phospholipids [83]. The highest ^•^OH scavenging percentages were exhibited by methanolic leaf extracts that can be ascribed to their high phenolic compositions. These phenolic compounds play a pivotal role in protection against oxidative damage leading to various degenerative diseases such as cardiac diseases, inflammatory diseases, and various kinds of malignancies [84]. The highest ^•^OH absorbing potential of methanolic leaf fractions was also supported by the antioxidant studies carried by Samak et al. [85]. The methanolic leaf extracts markedly scavenged the ^•^OH radicals and this activity was found to increase with the increase in the concentration of plant extracts. According to the results of our study, the order of IC_50_ values measured for the ^•^OH scavenging potential of methanolic leaf extracts from the three host trees was found as MLE PC > MLE UV > MLE JR (Table 1). Furthermore, the lowest IC_50_ value of 0.75 μg/mL was found in methanolic leaf extracts, most noticeably from the host *Populus ciliate* L., which further confirms that the methanolic leaf extract was highly effective in scavenging of ^•^OH radicals at very low concentrations and was also justified by their highest radical scavenging percentages.

The correlation found for samples from *Viscum album* L. and their host trees (*Populus ciliata* L. (PC), *Ulmus villosa* L. (UV), *Juglans regia* L. (JR)) confirms dependency of the host on the phenolic composition and antioxidant activity of *Viscum album* L. However, it can also be related to the epiphyte–host compatibility and *Viscum album’s* own phenolic production. Mistletoe (*Viscum album*) is hemi-parasitic species and grows on the stems and branches of a host plant, where it partially takes nutrients from the xylem, but also photosynthesizes its own carbohydrates [18]. The hydric stress generated by *Viscum album* L., as well as the excessive nitrogen uptake, could generate stress on the host plant; this could promote a high production of phenolic compounds in the host. The present study also showed higher total phenolic content (TPC), total flavonoid content (TFC), and enhanced antioxidant (FRAP, DPPH, SOR, ^•^OH) activities. Previous reports revealed that plants that grow in environments lacking nitrogen have higher secondary metabolites, in comparison to the ones that develop in a nitrogen-rich environment [86,87]; plants also increases the PAL (phenylalanine ammonium lyase) activity, the intermediate enzyme (branch point enzyme) for the synthesis of the phenolic compounds [87]. Previous reports also revealed excellent linear correlations between antioxidant activity and total phenolic content [88,89].

## 5. Conclusions

The results derived from the study revealed that the type of extraction solvent, host tree, and antioxidant assays affected the phytochemical content and antioxidant capacity of *Viscum album* L. The phytochemical yields of phenols and flavonoids were highest in ethanolic extracts. The same trend was seen in results obtained from other antioxidant assays, which signify that flavonoids and phenols strongly determine the antioxidant activities of the plant extract. Moreover, both phytochemical and antioxidant investigations suggested that leaf extracts from the host *Juglans regia* L. exhibited higher antioxidant activities and hence can be regarded as a potential source of antioxidants. Therefore, the crude leaf extracts of *Viscum album* L., irrespective of solvent used for extraction from the host *Juglans regia* L., warrants further research to unravel more pharmacological attributes based on its optimal biological potency in scavenging free radicals under in vitro systems.

## Figures and Tables

**Figure 1 plants-10-01191-f001:**
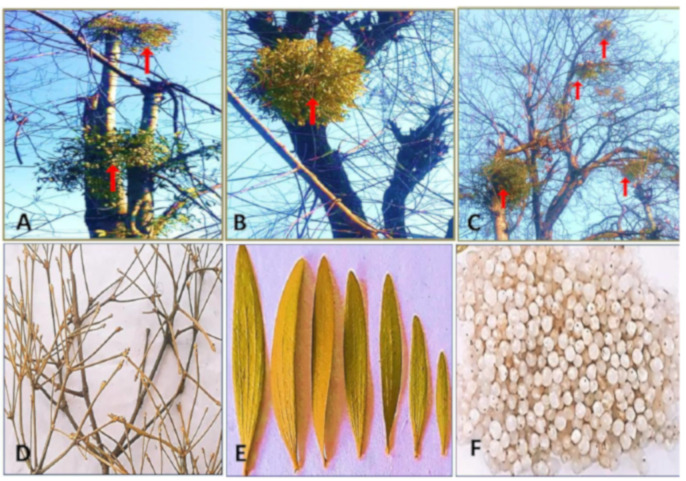
*Viscum album* L. harvested from three host trees (**A**) *Populus ciliata* L., (**B**) *Ulmus villosa* L., (**C**) *Juglans regia* L. Heavy infestation of epiphyte on the upper crown of the host plant is indicated by arrows. (**D**) Stem, (**E**) Leaves, (**F**) Berries.

**Figure 2 plants-10-01191-f002:**
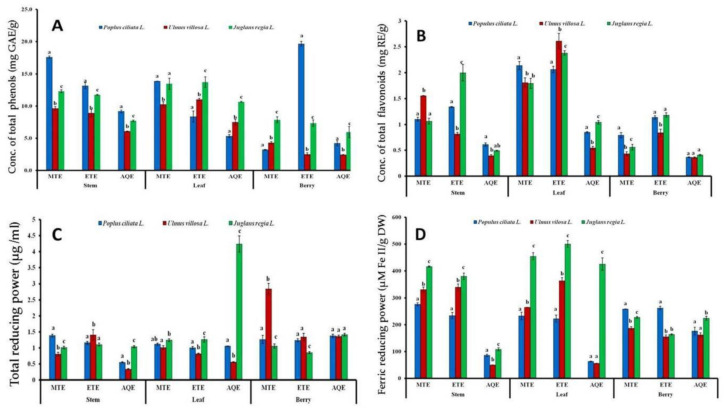
Variation in total phenolic contents (**A**), total flavonoid content (**B**), total reducing power (**C**), ferric reducing power antioxidant and (**D**) assays in *Viscum album* L. influenced by different hosts. Values are the mean ± SD of three replications. Sets of bars with different letters are significantly different (*p* < 0.05). MTE (methanolic extract), ETE (ethanolic extract), and AQE = (aqueous extract).

**Figure 3 plants-10-01191-f003:**
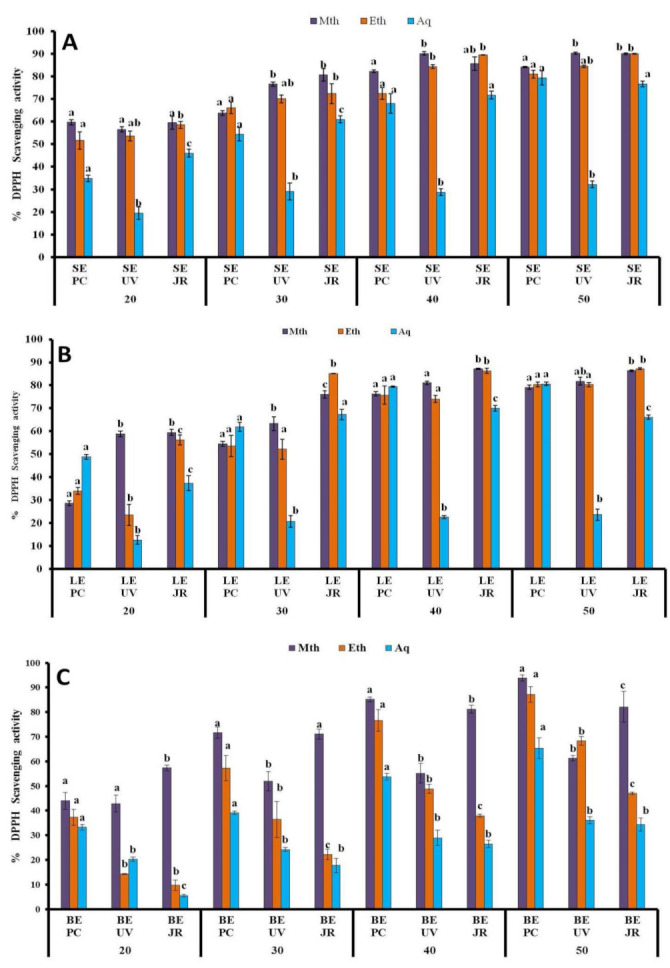
% DPPH scavenging potential of *Viscum album* L. stem (**A**), leaf (**B**), and berry (**C**) extracts at different concentrations (20 µL, 30 µL, 40 µL, and 50 µL) influenced by different hosts. Values are the mean ± SD of three replications. Bars with different letters are significantly different (*p* < 0.05). PC = *Populus ciliata* L., UV = *Ulmus villosa* L., JR = *Juglans regia* L.; SE= stem extract, LE = leaf extract, BE = berry extract.

**Figure 4 plants-10-01191-f004:**
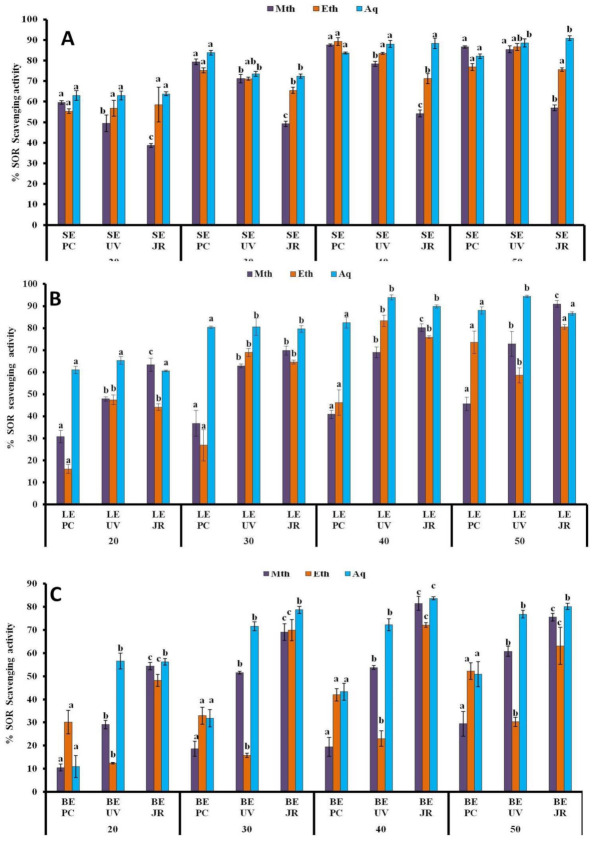
% SOR scavenging potential of *Viscum album* L. stem (**A**), leaf (**B**), and berry (**C**) extracts at different concentrations (20 µL, 30 µL, 40 µL, and 50 µL) influenced by different hosts. Values are the mean ± SD of three replications. Bars with different letters are significantly different (*p* < 0.05). PC = *Populus ciliata* L., UV = *Ulmus villosa* L., JR = *Juglans regia* L.; SE = stem extract, LE = leaf extract, BE = berry extract.

**Figure 5 plants-10-01191-f005:**
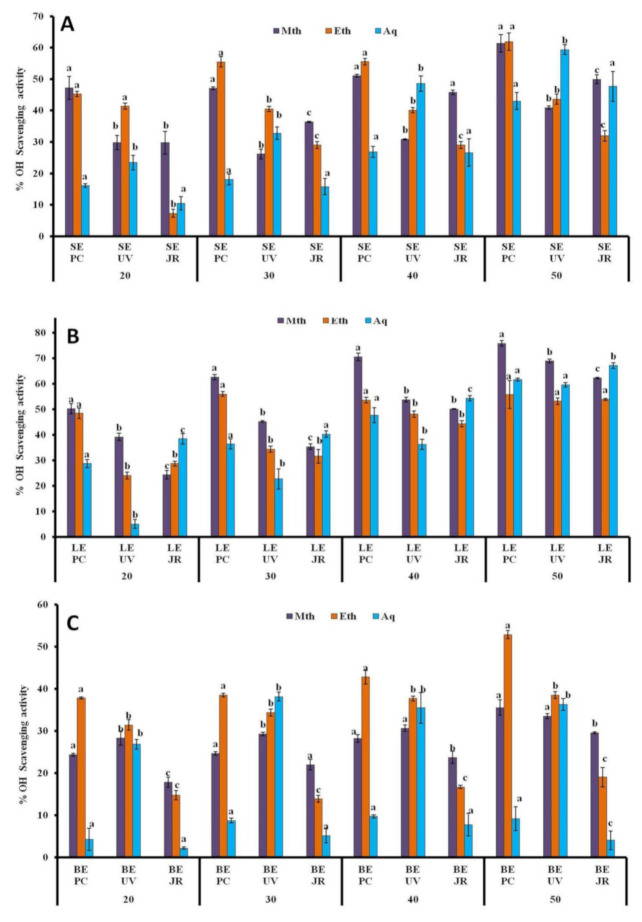
% OH scavenging activity of *Viscum album* L. stem (**A**), leaf (**B**), and berry (**C**) extracts at different concentrations (20 µL, 30 µL, 40 µL, and 50 µL) influenced by different hosts. Values are the mean ± SD of three replications. Bars with different letters are significantly different (*p* < 0.05). PC = *Populus ciliata* L., UV = *Ulmus villosa* L., JR = *Juglans regia* L.; SE = stem extract, LE = leaf extract, BE = berry extract.

**Figure 6 plants-10-01191-f006:**
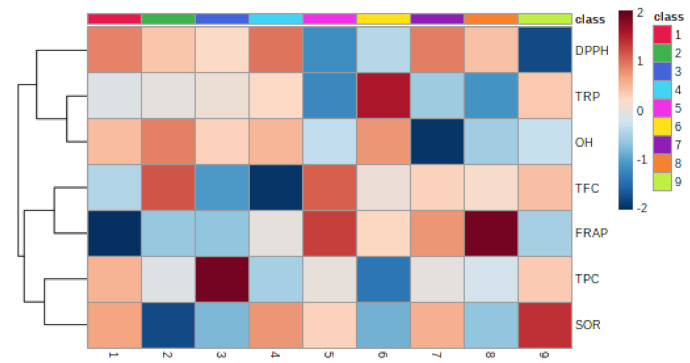
Correlation heatmap analysis among total phenols, flavonoids, and antioxidant activity of stem, leaf, and berry extracts of *Viscum album* L. collected from different hosts. *Populus ciliata* L. (PC): 1─PC stem; 2─PC leaf; 3─PC berry; *Ulmus villosa* L. (UV): 4─UV stem; 5─UV leaf; 6─UV berry; *Juglans regia* L. (JR): 7─JR stem; 8─JR leaf and 9─JR berry.

**Figure 7 plants-10-01191-f007:**
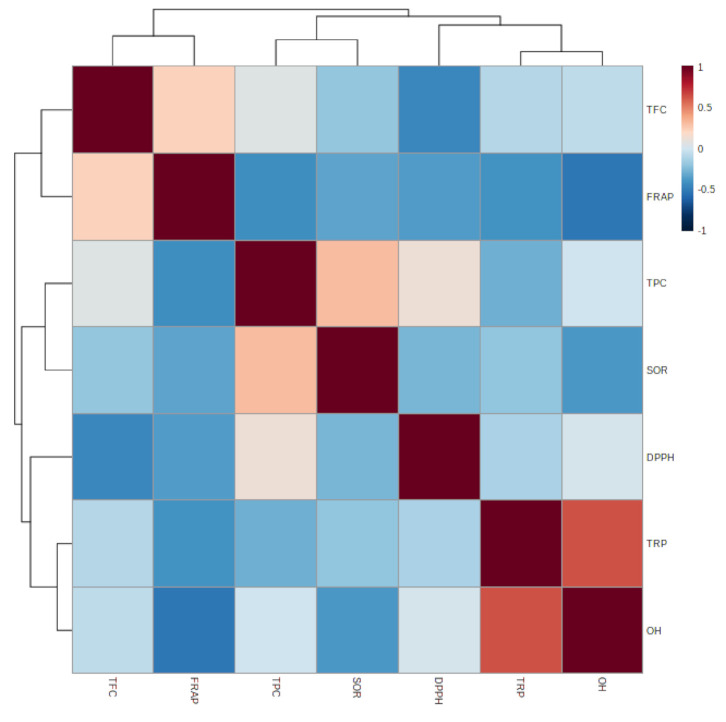
Pearson correlation coefficient for total phenolic content (TPC), total flavonoid content (TFC) and activities of antioxidants total reducing power (TRP), ferric reducing antioxidant power(FRAP), 1, 1-diphenyl 1-2-picryl-hydrazyl (DPPH), superoxide radical scavenging (SOR), and hydroxyl radical scavenging (OH-) of various parts of *Viscum album* L. collected from different hosts.

**Table 1 plants-10-01191-t001:** IC50 data for DPPH (A), SOR (B), and OH (C) scavenging activities of *Viscum album* L. collected from different hosts. * indicates the plant extracts with lowest IC50 values.

***Populus ciliata*** **L.**						
**DPPH (A)**	**IC50 M**	**R2**	**IC50 E**	**R2**	**IC50 A**	**R2**
Stem	0.0471 *	0.8954	0.617	0.97	1.879	0.9841
Leaf	1.9433	0.9122	1.823	0.9453	0.933	0.919
Berry	1.0492	0.9331	1.6369	0.9828	2.6944	0.9759
**SOR (B)**	**IC50 M**	**R2**	**IC50 E**	**R2**	**IC50 A**	**R2**
Stem	0.6698	0.7896	0.5783	0.5219	2.4171	0.5318
Leaf	44.6118	0.9937	2.2375	0.9641	0.8714	0.8318
Berry	7.796	0.9199	3.9222	0.9487	3.7001	0.9493
**OH (C)**	**IC50 M**	**R2**	**IC50 E**	**R2**	**IC50 A**	**R2**
Stem	1.5902 *	0.8799	2.1375	0.8029	5.183	0.8876
Leaf	0.7505 *	0.9653	0.7357 *	0.5177	3.0822	0.9835
Berry	3.9131	0.8461	8.3729	0.8455	47.3906	0.6563
***Ulmus villosa*** **L.**						
**DPPH (A)**	**IC50 M**	**R2**	**IC50 E**	**R2**	**IC50 A**	**R2**
Stem	0.0244 *	0.8664	0.3317	0.8845	8.4543	0.7898
Leaf	0.0424 *	0.8827	2.1093	0.9344	11.1628	0.8141
Berry	2.0207	0.9657	2.9606	0.9898	6.8233	0.9809
**SOR (B)**	**IC50 M**	**R2**	**IC50 E**	**R2**	**IC50 A**	**R2**
Stem	0.6678	0.9117	0.0955 *	0.9383	0.5958	0.9088
Leaf	0.874	0.9074	0.5531	0.1637	0.8363	0.8913
Berry	2.6237	0.8388	7.3755	0.9765	0.6575	0.8092
**OH (C)**	**IC50 M**	**R2**	**IC50 E**	**R2**	**IC50 A**	**R2**
Stem	7.2738	0.5988	15.9993	0.2713	3.2238	0.9908
Leaf	2.3186	0.9565	3.497	0.9731	3.577	0.9891
Berry	13.9943	0.9388	8.3498	0.9515	8.6177	0.4364
***Juglans regia*** **L.**						
**DPPH (A)**	**IC50 M**	**R2**	**IC50 E**	**R2**	**IC50 A**	**R2**
Stem	0.5006	0.8485	0.0251 *	0.906	1.1501	0.9539
Leaf	0.4733	0.8451	0.5471	0.6555	1.3503	0.5632
Berry	0.2172	0.8926	4.1274	0.9906	5.5522	0.9886
**SOR (B)**	**IC50 M**	**R2**	**IC50 E**	**R2**	**IC50 A**	**R2**
Stem	2.5363	0.9205	0.6081	0.99	0.4696	0.9386
Leaf	0.7908	0.7583	1.1484	0.9211	0.3061 *	0.9882
Berry	0.1589	0.7049	0.3438 *	0.3148	0.7142	0.6294
**OH (C)**	**IC50 M**	**R2**	**IC50 E**	**R2**	**IC50 A**	**R2**
Stem	3.8609	0.981	5.9722	0.6975	4.5285	0.9223
Leaf	3.0419	0.9972	3.6744	0.9521	1.9956	0.9281
Berry	46.197	0.961	24.0373	0.7804	57.8761	0.2045

**Table 2 plants-10-01191-t002:** Pearson correlations among total phenol, flavonoid, and antioxidant activity of *Viscum album* collected from different hosts.

	TFC	FRAP	TPC	SOR	DPPH	TRP	OH
**TFC**							
**FRAP**	0.231						
**TPC**	0.060	−0.414					
**SOR**	−0.192	−0.331	0.305				
**DPPH**	−0.448	−0.362	0.130	−0.265			
**TRP**	−0.093	−0.398	−0.282	−0.196	−0.122		
**OH**	−0.058	−0.520 *	−0.007	−0.375	0.018	0.636 **	

Note: * and ** indicates that correlation is significant at the 0.05 and 0.01 level.

## Data Availability

Not applicable.

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
