# Peer review of "Comparative Study on Phytochemical Profile and Antioxidant Activity of an Epiphyte, Viscum album L. (White Berry Mistletoe), Derived from Different Host Trees"

_plants, 2021, doi:10.3390/plants10061191_

Round 1
Reviewer 1 Report
The subject is interesting and the paper is well written. Some minor suggestions:
-A graphical scheme of sampling and extraction procedures should be added.
Introductive lines on update research of antioxidant properties (before describing in details TPC, TFC, FRAP, DPPH) and related references added such as:
Durazzo A. Study Approach of Antioxidant Properties in Foods: Update and Considerations. Foods. 02/2017; 6(3):17., DOI:10.3390/foods6030017.
Apak R. Current Issues in Antioxidant Measurement. J Agric Food Chem. 2019 Aug 21;67(33):9187-9202. doi: 10.1021/acs.jafc.9b03657. Epub 2019 Aug 6. PMID: 31259552.
Introductive lines also should be added to "3.1. Effect of host plants on phytochemical content " to better delineate the further subparagraphs.
A subparagraph including the correlations among different assays should be inserted.
Figure 3 and Table 1 should be better described in the text.
Author Response
Reviewer 1
The subject is interesting and the paper is well written. Some minor suggestions:
-A graphical scheme of sampling and extraction procedures should be added.
Response: (Done)
Introductive lines on update research of antioxidant properties (before describing in details TPC, TFC, FRAP, DPPH) and related references added such as:
Durazzo A. Study Approach of Antioxidant Properties in Foods: Update and Considerations. Foods. 02/2017; 6(3):17., DOI:10.3390/foods6030017.
Apak R. Current Issues in Antioxidant Measurement. J Agric Food Chem. 2019 Aug 21;67(33):9187-9202. doi: 10.1021/acs.jafc.9b03657. Epub 2019 Aug 6. PMID: 31259552.
Introductive lines also should be added to "3.1. Effect of host plants on phytochemical content " to better delineate the further subparagraphs.
Response: (Done) Paragraph including the recent studies conducted on effect of host plants has been added under section 3.1
A subparagraph including the correlations among different assays should be inserted.
Response: (Done)
Figure 3 and Table 1 should be better described in the text.
Response: (Done)
Reviewer 2 Report
As suggestion, I will recommend you to read with more attention the paper.
Author Response
As suggestion, I will recommend you to read with more attention the paper.
Reply: Manuscript has been substantially revised.
Reviewer 3 Report
In the paper " Comparative study on phytochemical profile and antioxidant activity of an epiphyte-Viscum album L. (white berry Mistletoe) derived from different host trees", the authors presented the biological potential of Viscum album L extracts. The study is very interesting and very well written. My biggest concern goes to the data analysis, the dependence between measured variables should be included in the manuscript. Therefore, my suggestion is major revision according to the following comments:
- Line 84. How were samples dried?
- Line 91. Ethanol and methanol concentrations used for extraction were not stated
- Line 94. How were the extraction conditions selected?
- The authors should analyze the correlations between measured variables (TPC, TFC, FRAP, DPPH, SOR, and OH). Currently, it is not clear is there a dependence between measured variables.
Author Response
In the paper " Comparative study on phytochemical profile and antioxidant activity of an epiphyte-Viscum album L. (white berry Mistletoe) derived from different host trees", the authors presented the biological potential of Viscum album L extracts. The study is very interesting and very well written. My biggest concern goes to the data analysis, the dependence between measured variables should be included in the manuscript. Therefore, my suggestion is major revision according to the following comments:
Line 84. How were samples dried?
Response: (Done).
Line 91. Ethanol and methanol concentrations used for extraction were not stated
Response: (Done).
Line 94. How were the extraction conditions selected?
Response: (Done).
The authors should analyze the correlations between measured variables (TPC, TFC, FRAP, DPPH, SOR, and OH). Currently, it is not clear is there a dependence between measured variables.
Response: (Done) The correlation analysis between assays has been included in paper
Graphical Abstract is attached
Round 2
Reviewer 3 Report
The authors put an effort and answered the comments. But there is still one important issue. Person correlations are linear correlations that can be applied if the data distribution is normal. How was the dana normality cheeked? Therefore, my suggestion is a minor revision prior to publication.
Author Response
Pearson's correlation does NOT assume normality. It is an estimate of the correlation between any two continuous random variables and is a consistent estimator under relatively general conditions. Even tests based on Pearson's correlation do not require normality if the samples are large enough because of the central limit theorem. When the variables are bivariate normal, Pearson's correlation provides a complete description of the association.